# *Limosilactobacillus reuteri* DSM 17938 Produce Bioactive Components during Formulation in Sucrose

**DOI:** 10.3390/microorganisms12102058

**Published:** 2024-10-12

**Authors:** Ludwig Ermann Lundberg, Manuel Mata Forsberg, James Lemanczyk, Eva Sverremark-Ekström, Corine Sandström, Stefan Roos, Sebastian Håkansson

**Affiliations:** 1Department of Molecular Sciences, Uppsala BioCenter, Swedish University of Agricultural Sciences, 750 07 Uppsala, Sweden; corine.sandstrom@slu.se (C.S.); stefan.roos@slu.se (S.R.); 2BioGaia AB, 112 27 Stockholm, Sweden; jl@biogaia.se (J.L.); sh@biogaia.se (S.H.); 3The Department of Molecular Biosciences, The Wenner-Gren Institute, Stockholm University, 114 18 Stockholm, Sweden; manuel.forsberg@su.se (M.M.F.); eva.sverremark@su.se (E.S.-E.); 4Division of Applied Microbiology, Department of Chemistry, Lund University, 221 00 Lund, Sweden

**Keywords:** lyoconversion, formulation, immunomodulation, exopolysaccharides, probiotics, metabolites

## Abstract

Improved efficacy of probiotics can be achieved by using different strategies, including the optimization of production parameters. The impact of fermentation parameters on bacterial physiology is a frequently investigated topic, but what happens during the formulation, i.e., the step where the lyoprotectants are added prior to freeze-drying, is less studied. In addition to this, the focus of process optimization has often been yield and stability, while effects on bioactivity have received less attention. In this work, we investigated different metabolic activities of the probiotic strain *Limosilactobacillus reuteri* DSM 17938 during formulation with the freeze-drying protectant sucrose. We discovered that the strain consumed large quantities of the added sucrose and produced an exopolysaccharide (EPS). Using NMR, we discovered that the produced EPS was a glucan with α-1,4 and α-1,6 glycosidic bonds, but also that other metabolites were produced. The conversion of the lyoprotectant is hereafter designated lyoconversion. By also analyzing the samples with GCMS, additional potential bioactive compounds could be detected. Among these were tryptamine, a ligand for the aryl hydrocarbon receptor, and glycerol, a precursor for the antimicrobial compound reuterin (3-hydroxypropionaldehyde). To exemplify the bioactivity potential of lyoconversion, lyoconverted samples as well as purified EPS were tested in a model for immunomodulation. Both lyoconverted samples and purified EPS induced higher expression levels of IL-10 (2 times) and IL-6 (4–6 times) in peripheral blood mononuclear cells than non-converted control samples. We further found that the initial cultivation of DSM 17938 with sucrose as a sugar substrate, instead of glucose, improved the ability to convert sucrose in the lyoprotectant into EPS and other metabolites. Lyoconversion did not affect the viability of the bacteria but was detrimental to freeze-drying survival, an issue that needs to be addressed in the future. In conclusion, we show that the metabolic activities of the bacteria during the formulation step can be used as a tool to alter the activity of the bacteria and thereby potentially improve probiotic efficacy.

## 1. Introduction

*Limosilactobacillus reuteri* DSM 17938 is a well-studied probiotic strain that has been shown to alleviate or prevent several disorders, including, but not limited to, infantile colic, constipation, and acute gastroenteritis [1,2,3,4]. Studies addressing probiotic function are growing in numbers, and the complexity of the mechanisms of action is enormous. In line with this is the search for effector molecules produced by *L. reuteri* DSM 17938, and we have recently published the effects of DSM 17938-derived extracellular membrane vesicles on cytokine responses in immune cells, the ability to reduce permeability in human epithelial cell monolayers, and antagonistic effects on the pain receptor TRPV1 [5]. Proteomic analysis revealed that one of the most abundant proteins found on the membrane vesicles and the bacterial cell surface is a glucansucrase that produces an exopolysaccharide (EPS). It has previously been described that some strains of *L. reuteri* produce EPSs of the glucan and fructan types during cultivation, given that the substrate sucrose is available in the culture medium [6,7].

EPSs are commonly produced by numerous bacterial species, including commensal, probiotic, and pathogenic bacteria, and different types of functions have been described. EPS from *Streptococcus pneumoniae* is considered to be a virulence factor that facilitates colonization [8]. *L. reuteri* DSM 17938-derived EPS has been shown to inhibit the adhesion of enterotoxin-producing *E. coli* (ETEC) to intestinal IPEC-1 cells [9]. EPSs can also exert immunomodulatory effects [10]. EPSs are divided into two different groups: homopolysaccharides (HoPSs) and heteropolysaccharides (HePSs) [11]. EPSs, such as the glucan named reuteran produced by *L. reuteri* DSM 17938 [7], and consisting of only one type of sugar moieties, belong to the HoPS group and have commonly been associated with food applications. However, HoPSs have been less studied in terms of their molecular structures, in contrast to more complex HePSs, which have also been suggested to have prebiotic properties [12].

The industrial production of probiotics is accompanied by several challenges and differs substantially from lab-scale production [13]. One of the main differences between the lab scale and the industrial scale is the holding times, i.e., the time between two steps in the production process, which can amount to hours simply because of the large volumes handled. Holding times mean that the production process is halted but does not necessarily mean that the bacterial metabolism is inactive during this period. Our aim was to combine the knowledge of *L. reuteri* DSM 17938, which carries a surface-anchored glucansucrase that produces a glucan from sucrose [14], with a process design that includes a defined holding time during the formulation step. By allowing EPS production to occur during the formulation step rather than during cultivation, the produced EPS remains in the product, instead of being lost to the spent culture medium during downstream processing. Therefore, the purpose of this study was to evaluate if EPSs and potentially other metabolites are produced during the holding time with this approach. Additionally, we investigated if any such conversions lead to changes in survival throughout the production process and impact immunomodulatory activity.

## 2. Materials and Methods

### 2.1. Fermentation, Lyoconversion, and Freeze-Drying

*Limosilactobacillus reuteri* DSM 17938 was provided by BioGaia AB [15]. All substrates were autoclaved for 15 min at 121 °C. The strain was inoculated from frozen stocks into 10 mL De Man, Rogosa, and Sharpe (MRS) broth (Oxoid) and cultivated for 16 h at 37 °C without agitation. The next day, 4 mL was inoculated into 400 mL of a *Lactobacillus*-carrying medium (LCM) [16] with 1% glucose or sucrose and incubated for 24 h at 37 °C without agitation. Thereafter, the cultures were centrifuged at 4000× *g* for 10 min, and the bacterial pellet was resuspended in 40 mL of 10% sucrose. Half of the suspension was transferred to 2 mL vials and directly frozen at −50 °C, while the remainder was allowed to incubate in the sucrose suspension for 24 h at room temperature and frozen at −50 °C afterward in the same manner as the direct-frozen samples. The samples were freeze-dried with a Christ Epsilon 2-6D LSCplus according to the protocol in Appendix A. A description of the samples can be seen in Figure 1.

The pH was measured using a Mettler Toledo pH meter. Four replicates of freeze-dried samples were resuspended in 2 mL water, after which pH was measured. CFU measurements after lyoconversion as well as freeze-drying were performed by plating on MRS agar plates (Merck) and anaerobic incubation at 37 °C for 48 h. The freeze-dried bacteria were resuspended in 2 mL PBS before the CFU analysis.

A schematic overview of samples and methods used in this paper is presented in Figure 1.

### 2.2. NMR Spectroscopy

Each freeze-dried sample was rehydrated in 2 mL D_2_O followed by centrifugation at 4000× *g* for 15 min. Filtered supernatants (0.45 µm filter) were freeze-dried in a Coolsafe freeze dryer (Scanvac, Lynge, Denmark), while the cell pellets were discarded. The freeze-dried supernatant was resuspended in 700 µL D_2_O, and the procedure was repeated five times. Prior to the NMR analysis, the samples were suspended in 588 µL D_2_O together with the internal standard trimethylsilyl-d4-propionic acid (TSP, 12 µL, 5.8 mmol/L). Each sample solution (600 µL) was transferred into 5 mm NMR tubes (Bruker). The NMR spectra were recorded on a Bruker Avance III 600 MHz spectrometer with a 5 mm broadband observe detection Smartprobe equipped with z-gradient. TSP (δ_H_ 0.00 and δ_C_ 0.00) was used as an internal reference for chemical shifts and for quantification. TopSpin version 4.0.9 (Bruker) was used for data processing. The one- and two-dimensional NMR experiments including 1H, COSY, TOCSY, and HSQC were recorded using standard pulse sequences from the Bruker library. A mixing time of 120 ms was used for the TOCSY experiments. Additionally, 1D diffusion-edited NMR spectra that allow for the suppression of signals from small molecules were recorded using the BPP-LED pulse sequence ledbpgp2s1d from the Bruker library.

### 2.3. Extraction of Exopolysaccharides

A vial of freeze-dried sucrose-grown bacteria incubated for 24 h was resuspended in 2 mL water and centrifuged at 4000× *g* for 15 min. Pellets were discarded, and proteins in the supernatant were precipitated for 15 min at 2 °C by adding trichloroacetic acid at 20% final concentration. After 15 min of incubation, the suspension was centrifuged at 25,000× *g* for 30 min at 4 °C. The pellets were removed, and one equivalent volume of acetone was added to the supernatant, after which it was incubated at 2 °C for 24 h. Then, the suspension was centrifuged at 25,000× *g* for 30 min, at 4 °C, and the pellet with EPSs was collected, suspended in 1 mL water, and dried in a Coolsafe freeze dryer (Scanvac, Lynge, Denmark).

### 2.4. Cytokine Secretion in Peripheral Blood Mononuclear Cells

Venous blood was extracted from healthy anonymous adult volunteers and diluted 1:1 in RPMI-1640 with 20 mM HEPES (HyClone Laboratories, Inc., Logan, UT, USA). Ficoll-Hypaque (GE Healthcare Bio-Sciences AB, Uppsala, Sweden) gradient separation was used to extract peripheral blood mononuclear cells (PBMCs) that were then washed, suspended, and frozen in a medium containing 40% RPMI-1640, 50% fetal bovine serum (FBS) (Sigma-Aldrich, Saint Louis, MO, USA) and 10% DMSO (Sigma-Aldrich). The PBMCs were cooled gradually at −80 °C in freezing containers (Mr. Frosty, Nalgene Cryo 1 °C; Nalge CO., Rochester, NY, USA) and stored in liquid nitrogen before being thawed and cultured for the cytokine response assays. Before being seeded in flat-bottom 96-well plates, the cells were stained with Trypan blue and counted in a 40× light microscope. The seeding concentration was 2.5 × 10^5^/well, and the cells were incubated for 48 h at 37 °C with 5% CO_2_. Stimulation experiments were performed with a multiplicity of bacteria (MOB) equivalent of 100:1 and 10:1. Cell supernatants were collected and frozen. The secreted cytokine levels were quantified using sandwich ELISA kits (MabTech AB, Stockholm, Sweden) following the manufacturer’s instructions. The absorbance was measured at 405 nm in a microplate reader (Molecular Devices Corp. San Jose, CA, USA), and the results were analyzed using SoftMax Pro 5.2 rev C (Molecular Devices Corp.) and subsequently statistically analyzed in Prism GraphPad 9.0 (GraphPad Software Inc., La Jolla, CA, USA).

### 2.5. High-Performance Liquid Chromatography (HPLC)

Volatile fatty acids were analyzed by high-performance liquid chromatography (HPLC). Agilent 1100 Series, with a refractive index detector and an ion-exclusion column (Rezex ROA–Organic Acid H+, 300 × 7.80 mm, Phenomenex, Torrance, CA, USA). The mobile phase used was 5 mM H_2_SO_4_ with a flow rate of 0.6 mL min^−1^. Samples were prepared by pipetting 700 µL of the sample into a microcentrifuge tube, and then 70 µL of 5 M H_2_SO_4_ was added and mixed with the sample before being centrifuged at 14,000× *g* for 15 min. The supernatant was filtered through a 0.2 µm syringe filter into an HPLC glass vial. Two internal references containing approximately 0.5 g/L and 4 g/L of the following fatty acids were used: acetate, propionate, butyrate, isobutyrate, valerate, and isovalerate. 2,3-butanediol was also used as an internal reference for verification.

### 2.6. Gas Chromatography–Mass Spectrophotometry

Metabolic profiling by GC-MS was performed at the Swedish Metabolomics Center in Umeå, Sweden.

Sample preparation was performed as described by [17]. In brief, 950 µL of the extraction buffer (90/10 *v*/*v* methanol–water) including internal standards for GC-MS (L-proline-13C5, alpha-ketoglutarate-13C4, myristic acid-13C3, cholesterol-D7 (Cil (Andover, MA, USA)), succinic acid-D4, salicylic acid-D6, L-glutamic acid-13C5,15N, putrescine-D4, hexadecanoic acid-13C4, D-glucose-13C6, D-sucrose-13C12 (Sigma, St. Louis, MO, USA)) was added to 50 µL of the resuspended, centrifuged, and filtered (0.2 μm) freeze-drying supernatant. Said mixes were shaken at 30 Hz for 2 min in a mixer mill, and proteins were precipitated at 4 °C on ice. Samples were then centrifuged at 4 °C, 14,000 rpm, for 10 min. Briefly, 10 µL of the supernatant was transferred to microvials and evaporated to dryness in a Speed-Vac concentrator. Solvents were evaporated, and the samples were stored at −80 °C until analysis. Small aliquots of the remaining supernatants were pooled and used as quality control (QC) samples. The samples were analyzed in batches according to a randomized run order on GC-MS.

Derivatization and GCMS analysis were performed as described previously. Briefly, 1 μL of the derivatized sample was injected in split mode (1:10) with a L-PAL3 autosampler (CTC Analytics AG, Zwingen, Switzerland) into an Agilent 7890B gas chromatograph equipped with a 10 m × 0.18 mm fused silica capillary column with a chemically bonded 0.18 μm Rxi-5 Sil MS stationary phase (Restek Corporation, Bellefonte, PA, USA). The injector temperature was 270 °C, the purge flow rate was 20 mL min^−1^, and the purge was turned on after 60 s. The gas-flow rate through the column was 1 mL min^−1^, and the column temperature was held at 70 °C for 2 min and then increased by 40 °C min^−1^ to 320 °C and held there for 2 min. The column effluent was introduced into the ion source of a Pegasus BT time-of-flight mass spectrometer, GC/TOFMS (Leco Corp., St Joseph, MI, USA). The transfer line and the ion source temperatures were 250 °C and 200 °C, respectively. Ions were generated by a 70 eV electron beam at an ionization current of 2.0 mA, and 30 spectra s^−1^ were recorded in the mass range *m*/*z* 50–800. The acceleration voltage was turned on after a solvent delay of 150 s. The detector voltage was 1800–2300 V.

All non-processed MS files from the metabolic analysis were exported from the ChromaTOF software (version 2.32) in NetCDF format to MATLAB^®^ R2022a (MathWorks, Natick, MA, USA), where all data of pre-treatment procedures, such as base-line correction, chromatogram alignment, data compression, and multivariate curve resolution, were extracted. The extracted mass spectra were identified by comparisons of their retention indices and mass spectra with retention time indices as well as mass spectra indicated in libraries [18]. Mass spectra and retention index comparisons were performed using NIST MS 2.2 software. Annotation of mass spectra was based on reverse and forward searches in the library. Masses and ratios between masses indicative of a derivatized metabolite were especially annotated. The mass spectrum with the highest probability indicative of a metabolite was determined, and the retention index between the sample and library for the suggested metabolite was ±5 (usually less than 3); the deconvoluted “peak” was annotated as the identification of a metabolite.

### 2.7. Statistical Analysis

Prism GraphPad version 9.0 (GraphPad Software, Boston, MA, USA) was used in all statistical data analyses. ANOVA and Tukey’s post hoc statistical analyses were performed for the pH measurements and CFU measurements after lyoconversion. A paired *t*-test was used to compare the CFU values after freeze-drying.

Friedman’s test with Dunn’s multiple-comparison tests was performed for the cytokine secretion evoked by formulation. Wilcoxon matched-pair signed-rank test was performed for the cytokine secretion evoked by the extracted EPS. The GCMS data were analyzed using ANOVA with Tukey’s post hoc multiple-comparison test. Data from six separate vials from each biological replicate were averaged, and data from three vials passed the normality test (Shapiro–Wilk). 

## 3. Results

### 3.1. L. reuteri DSM 17938 Produce an α-Glucan during Formulation

As the first step, the ability of *L. reuteri* DSM 17938 to produce EPSs during the formulation step was analyzed. After preparing the samples according to the schematic overview in Figure 1, we analyzed the content of sucrose and glycosidic bonds using NMR. The ^1^H-spectra of the two direct-frozen samples (sucrose-DF and glucose-DF) showed dominant signals from sucrose, while the spectra of the two samples incubated in the lyoprotectant at room temperature for 24 h (sucrose-RT and glucose-RT) showed a clear decrease in the sucrose concentration (as shown by the decrease in the intensity of the ^1^H signal of H1 Glc of sucrose at δ 5.41 ppm; Figure 2). Anomeric signals at δ 5.234 and 4.647 ppm for α-Glc revealed a release of free glucose and broad signals at δ 4.97 and δ 5.35 ppm indicated a production of α-(1-4) and α-(1-6) glucan. The process during which the bacteria converted sucrose during the formulation was denoted lyoconversion.

The α-(1-4) anomeric ^1^H signal splits into three overlapping broad signals at δ 5.34, 5.33, and 5.30 ppm, indicating the presence of α-D-Glcp-(1-4) residues in at least three significantly different structural fragments. The α-(1-6) anomeric ^1^H signal splits into two overlapping broad signals in sucrose-RT and three broad signals in glucose-RT, indicating different α-D-Glcp-(1-6) structural elements. These structures were confirmed through the assignment of ^1^H and ^13^C chemical shifts by 2D-NMR and comparison with previously published NMR data on similar polysaccharides as well as on starch and glycogen [12,19,20,21,22,23].

### 3.2. The Sugar Substrate Impacts Outcome of Lyoconversion

As a next step, we investigated whether the cultivation of *L. reuteri* with sucrose as the sugar substrate induced the enzyme needed for lyoconversion of sucrose in the formulation or not. The interpretation of the NMR spectra indicated that, in the sucrose-RT samples, approximately 68% of the glucose residues were 1-4-linked, and 32% were 1-6-linked. In glucose-RT, 73% were 1-4-linked and 27% were 1-6-linked (Figure 3). Another interesting observation was that the glucose level increased in the sucrose-RT samples, indicating that the polymerization was slower than the hydrolysis of sucrose. There were several unknown peaks that had different appearances depending on the sugar substrate, for example, the peaks between 0.5 and 2.5 ppm.

The complexity of the NMR spectra of glucose-RT and sucrose-RT samples indicated that many other metabolites were produced during lyoconversion. The metabolite 2,3-butanediol with characteristic ^1^H/^13^C chemical shifts at δ, 1.12/19.5 ppm (CH_3_ group) and δ 3.71 /73.6 ppm (CH group) was identified in the NMR spectra of both glucose-RT and sucrose-RT samples but was produced in higher quantity in the latter samples (Figure 2). 

The samples were analyzed by GCMS to help decipher what other metabolites were produced during the lyoconversion of sucrose. In this analysis, the relative concentrations of sucrose, glucose, and fructose were determined (Figure 4), showing that there were large differences between samples and that the efficiency of lyoconversion is dependent on the sugar in the growth substrate. The use of sucrose in the growth substrate resulted in higher lyoconversion of sucrose to glucose and fructose, compared to the use of glucose as a sugar growth substrate (Figure 4). Notably, the size of the sucrose peaks in the NMR spectra appeared to be of the same magnitude, highlighting a discrepancy between the GCMS and NMR results. However, an increased glucose concentration in sucrose-grown RT samples was observed by using both NMR and GCMS.

The GCMS analysis also revealed that lyoconversion greatly affected the syntheses of many different compounds with potential bioactivities and that the samples differed substantially (Figure 5A). GCMS identified 43 different metabolites, and 35 of them differed depending on lyoconversion (DF vs. RT). Sixteen of the detected metabolites were sugars, twelve were amino acids, and six were sugar alcohols. A full comparison of the 43 metabolites is shown in Appendix A. Relative concentrations of selected metabolites are displayed in Figure 5B. For the most part, concentrations were higher in the sucrose-RT samples. For example, malonic acid, glycerol, glyceraldehyde, and tryptamine were significantly increased (Figure 5B). Not all metabolites were detected at higher concentrations in the sucrose-RT samples, as exemplified by mevalonic acid (Figure 5B), where glucose-RT samples had elevated levels compared to sucrose-RT samples. Sorbitol was also increased in the lyoconverted samples (Appendix A). However, mannitol has the same retention time as sorbitol in the GCMS setup used, and therefore the higher levels of sorbitol may indicate the production of mannitol, which is a likely scenario during sucrose metabolism leading to the utilization of fructose as an electron acceptor [24].

### 3.3. Lyoconversion Decreases pH, Increases Organic Acid Production, and Reduces Freeze-Drying Survival

Since sucrose could be converted to both an EPS and free monosaccharides, we investigated if the bacteria could further metabolize the sugars to organic acids and thereby reduce the pH. Such metabolism may have an impact on bacterial survival through the lyoconversion process as well as after freeze-drying, which was also investigated. Indeed, pH was decreased as a result of lyoconversion, and sucrose-RT samples had the lowest pH (Figure 6A). The difference was greater in the sucrose-grown samples with an average pH of 3.34 in the sucrose-RT and pH 4.16 in sucrose-DF. A smaller difference was observed in the glucose-grown samples, with a pH of 3.73 in glucose-RT and a pH of 4.10 in glucose-DF. Furthermore, it was also shown that lactate and acetate were produced during the lyoconversion. The sucrose-RT bacteria produced the most organic acids, as expected (Figure 6B). In the sucrose-RT samples, the amounts of lactate and acetate were more than six times higher than in the sucrose-DF samples (1.06 g/L to 7.07 g/L and 1.06 g/L to 6.40 g/L, respectively). In the glucose-RT samples, there was also a higher amount of lactate (2.11 g/L to 3.44 g/L), and acetate produced (0.35 g/L to 1.31 g/L). There was no direct effect of lyoconversion on the survival of the concentrated cells prior to freeze-drying (Figure 6C). As proof of principle, we evaluated the freeze-drying survival in the samples with highest degree of lyoconversion (grown in sucrose), and it was found that survival was greatly reduced in the RT samples (Figure 6D). By comparing the CFU values before and after freeze drying (Figure 6C,D), the survival in the sucrose-DF samples were found to be 25%, whereas the survival in the sucrose-RT samples were 0.14%.

### 3.4. Lyoconverted Samples Increase Cytokine Secretion from PBMC

As some types of EPSs [25] have been shown to be immunogenic, the secretion of IL-6 and IL-10 was evaluated in PBMCs exposed to components from cell-free formulations. IL-6 and IL-10 were selected as representative cytokines for immune activation and anti-inflammatory activity, respectively [26,27]. The lyoconverted samples induced higher levels of interleukin secretion in PBMCs, showing that the content of the converted samples is immunomodulatory (Figure 7).

Regardless of the sugar growth substrate, we observed a higher IL-6 secretion as a response to RT samples compared to DF samples. In the analysis of IL-10, only the glucose-grown samples differed between DF and RT.

Finally, we extracted the EPS from lyoconverted samples and evaluated its immunogenic effect. Both IL-6 and IL-10 production were induced by the EPS. The IL-10 assay was performed by using the same MOB equivalent as in the lyoconverted samples and IL-10 was slightly less induced for EPS than sucrose-RT. The IL-6 induction was also partly attributable to the EPS, although the response to sucrose-RT was stronger (MOB equivalent to 10, Figure 7) as compared to the extracted EPS (MOB equivalent of 100, Figure 8). This indicates that other components produced during lyoconversion are also immunomodulatory.

## 4. Discussion

The consensus is that lyoprotectants used to stabilize probiotics are stable during freezing and drying [28]. However, since certain probiotic strains can be metabolically active without growing and lyoprotectants often contain sugars, we hypothesized that lyoprotectants could be metabolized during production. Industrial processes commonly involve holding times [13], constituting a window of opportunity for the bacteria to continue metabolic activities and thus alter the composition of the formulation. While this may appear problematic, it could enhance probiotic efficacy by the production of bioactives. Since some strains of *L. reuteri* carry a cell-surface-located sucrase [29], we refined our hypothesis to suggest that *L. reuteri* DSM 17938 could convert sucrose to exopolysaccharides (EPSs) and other metabolites. Through NMR and GCMS, we found that *L. reuteri* DSM 17938 converts sucrose into an α-(1-4), α-(1-6)-glucan, as well as lactate and acetate, reducing the pH of the formulated cell suspension. As mentioned, DSM 17938 is genetically equipped with a glucansucrase and has been reported to produce glucans before [30]. However, to the best of our knowledge, DSM 17938 has never been reported to produce any fructans, and through genomic assessment, we found that it has a truncated levansucrase gene, rendering it unable to produce fructans. We found many other potentially bioactive compounds produced during the process. Thus, the metabolic conversion of the lyoprotectant, a phenomenon we term lyoconversion, can occur and impact the properties of the freeze-dried probiotic bacteria. The discovery of lyoconversion may aid in improving the bioactivity of existing probiotic products but also partly explain issues with freeze-drying survival and storage stability.

Normally, a probiotic product is assessed and evaluated with respect to viable cell count and storage stability, and we show here that the lyoconversion had no effect on CFU or survival after freezing but had a detrimental effect on freeze-drying survival with less than 1% survival. This is most likely due to the consumption of the non-reducing disaccharide sucrose and the production of acids and the reducing sugars fructose and glucose, as well as sugar alcohol mannitol. With the increased concentration of the reducing sugars fructose and glucose, as well as mannitol, there will be increased Maillard reactions having detrimental effects on long-term stability [31]. A conclusion that can be drawn is that either sucrose should be replaced with a non-convertible protectant, or that the lyoconversion needs to be compensated for by the addition of a second helping of a stabilizing agent. For example, a process using sucrose and lyoconversion, followed by another addition of sucrose prior to freezing, could potentially result in a more stable probiotic product. Other routes for increasing the freeze-drying survival of lyoconverted probiotics would be to introduce a more complex lyoprotectant composition, e.g., adding a buffering compound, employing encapsulation techniques, or incorporating annealing procedures into the freeze-drying protocol [32,33]. However, if a more complex lyoprotectant is used, a potential lyoconversion of the newly introduced substances needs to be investigated. Contrastingly, lyoconversion may play a role in the emerging field of postbiotics, where the survival and stability of the microorganisms do not matter. How lyoconversion may potentiate postbiotic products is an interesting topic for future investigations.

While the effects of lyoconversion on stability are potentially problematic, we wanted to investigate the potential upside of the phenomenon. We showed how lyoconverted samples increased IL-6 and IL-10 responses from PBMCs, potentially paving the way for alternative production strategies with the goal of increased bioactivity. There were multiple interesting compounds being produced during lyoconversion, including substantial amounts of EPSs. We showed that EPSs are partly responsible for the immunomodulatory properties of lyoconverted DSM 17938, albeit not the sole bioactive component. When Kšonžeková and colleagues evaluated EPSs derived from *L. reuteri* ATCC 55730, the mother strain of DSM 17938, they showed that the EPSs could inhibit adherence to cultivated epithelial cells and reduce the proinflammatory effects of enterotoxigenic *Escherichia coli* [9].

In addition, we also detected 2,3-butanediol, and the concentration was highest in the sucrose-RT samples. The production of 2,3-butanediol may be a pH homeostasis-associated response to the lower pH observed in the lyoconverted samples, where the acetoin pathway is utilized, thereby avoiding the acidic end-products lactate and acetate. Similar effects have been described in *Lactiplantibacillus plantarum*, although the conversion of pyruvate for that species was halted at acetoin [34]. The conversion of pyruvate into 2,3-butanediol is interesting from a chemical perspective as it has multiple applications in a variety of different fields, e.g., plasticizers, foods, and pharmaceuticals [35]. The fact that 2,3-butanediol is used as a plasticizer is interesting from a stability point of view and might shed some light on the accompanying long-term stability issues associated with lyoconversion. Besides water, the presence of other plasticizers reduces the glass transition temperature (Tg′), which could give rise to downstream issues with stability [36,37].

The GCMS analysis also revealed that the lyoconverted samples contained tryptamine, which is a potent neuroendocrine and immunological signal molecule. Tryptamine production by *L. reuteri* DSM 17938 has previously been demonstrated [38], and the compound has been attributed several important functions and is an aryl hydrocarbon receptor (AhR) ligand involved in the finetuning of mucosal reactivity [39]. Intestinal tryptamine is also known to induce serotonin production by enterochromaffin cells. Serotonin is a neurotransmitter that has immunological, neurological, and gastrointestinal motility-modulating effects [40]. However, the absolute concentration of tryptamine in the lyoconverted sucrose sample appeared to be low, as they were not detected in the NMR analysis. Nevertheless, this observation raises the question of whether lyoconversion may affect tryptophan metabolism and if it can be used to deliver tryptophan catabolites with importance for intestinal homeostasis. Glycerol was also detected in the sucrose-RT samples, which is of specific interest as it is involved in the production of the antimicrobial compound 3-hydroxypropionaldehyde (reuterin) by *L. reuteri* [41]. An endogenous production of glycerol in *L. reuteri* has, to our knowledge, not been described before and would be interesting for further research. On the other hand, we could not find any 3-hydroxypropionaldehyde in the samples, which may indicate that what was detected as glycerol may be glycerol-3-phosphate derived from the lipid membrane of the bacteria, since these compounds were undistinguishable using GCMS. Further, malonic and mevalonic acids were produced, which may indicate the production of hopanoids that aid in fatty acid synthesis via the isoprenoid pathway [42]. The untargeted GCMS methodology used herein does not quantify the absolute concentrations of the detected compounds, but the results can be used as a basis for future studies. Studies addressing quantities and function of the produced compounds, for instance, by targeted GCMS in combination with a suitable preclinical model, are warranted.

Finally, several studies concerning production parameters during fermentation and their impact on the physiology of the bacteria have been published in the last decade [43,44,45,46], but to the best of our knowledge, lyoconversion has not previously been described. With the discovery of bioactive components produced by converting the common lyoprotectant sucrose, we anticipate that optimized lyoconversion could be a tool for improving probiotic functionality. Future research may address overcoming the poor freeze-drying survival in order for lyoconversion to be used in the production probiotics, or for the concept to be used as a means of producing potent postbiotics.

## Figures and Tables

**Figure 1 microorganisms-12-02058-f001:**
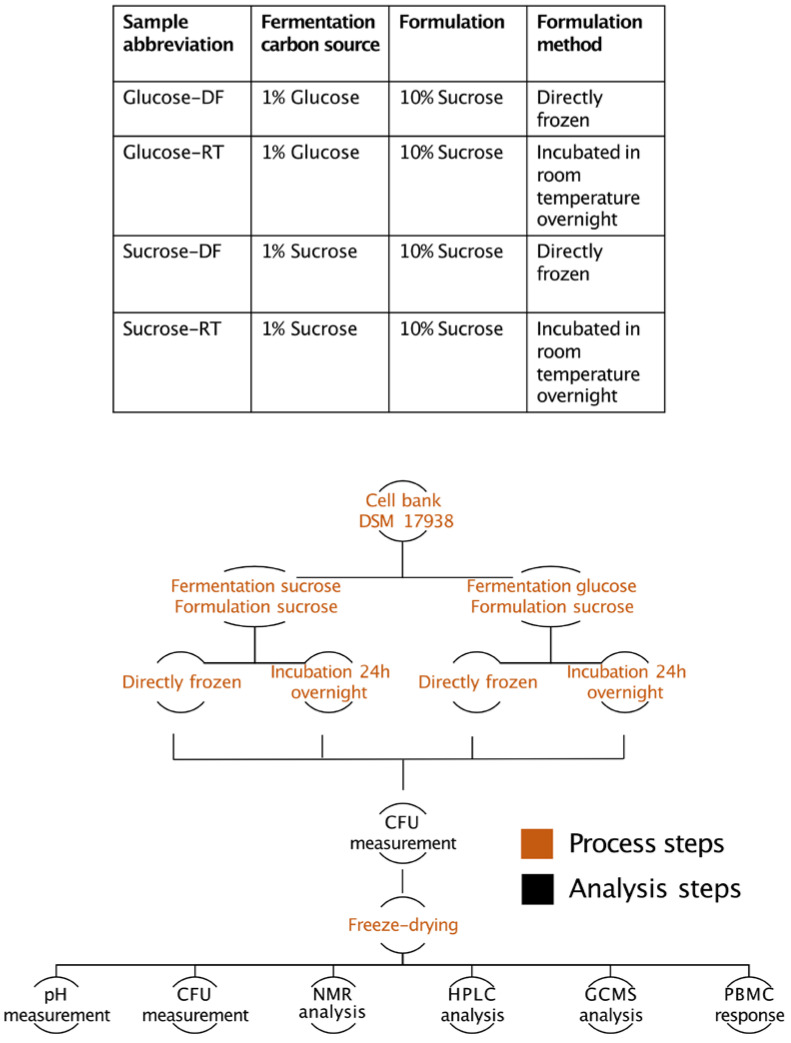
Schematic overview of samples and methods used in this work.

**Figure 2 microorganisms-12-02058-f002:**
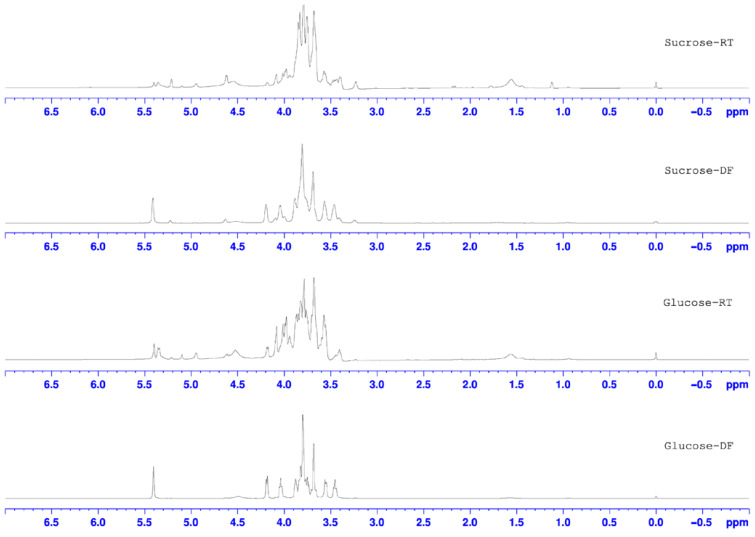
1D ^1^H NMR spectra recorded at 333 K of freeze-drying supernatants of sucrose- and glucose-grown bacteria, either incubated in lyoprotectant at room temperature overnight (RT) or directly frozen (DF).

**Figure 3 microorganisms-12-02058-f003:**
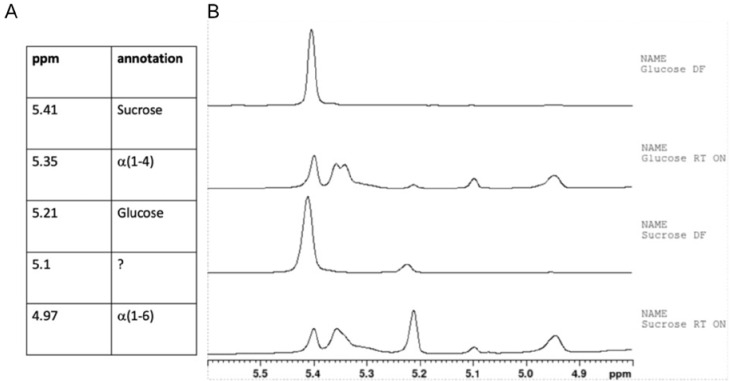
Anomeric region of the 1D 1H NMR spectra. Sucrose- and glucose-grown bacteria were incubated at room temperature overnight (RT) or directly frozen (DF). The chemical shifts of the anomeric proton signal of glucose in sucrose, in α-(1-4)-linked EPSs, free α-Glc, and α-(1-6)-linked EPSs are indicated in the table (**A**). The NMR spectra are reported in (**B**). ? indicates a peak that could not be annotated.

**Figure 4 microorganisms-12-02058-f004:**
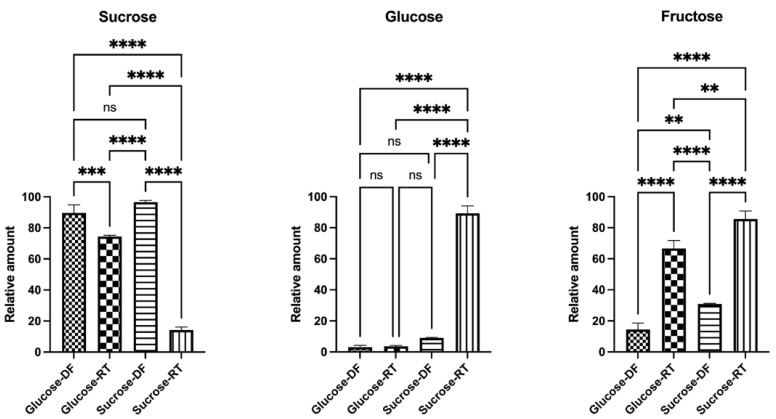
The relative amount of sucrose, glucose, and fructose in the lyoconverted samples determined by GCMS. Significance levels used were ** *p* < 0.01; *** *p* < 0.001, **** *p* < 0.0001, ns *p* > 0.05. Prism GraphPad version 9.0 was used for statistical analysis.

**Figure 5 microorganisms-12-02058-f005:**
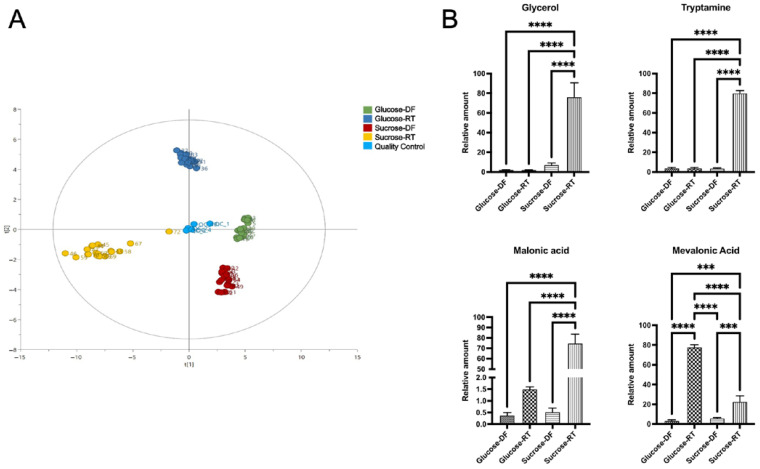
GCMS metabolomic data displayed as a PCA plot with sample distributions and relative concentrations of selected metabolites. (**A**) Sample distributions show four distinct clusters corresponding to the sample treatments. (**B**) Briefly, the sucrose-grown lyoconverted samples had higher relative concentrations of most of the identified metabolites including glycerol, glyceraldehyde, tryptamine, and malonic acid, whereas glucose-grown lyoconverted samples had increased relative concentrations of mevalonic acid. Significance levels used were *** *p* < 0.001, **** *p* < 0.0001. Prism GraphPad version 9.0 was used for statistical analysis.

**Figure 6 microorganisms-12-02058-f006:**
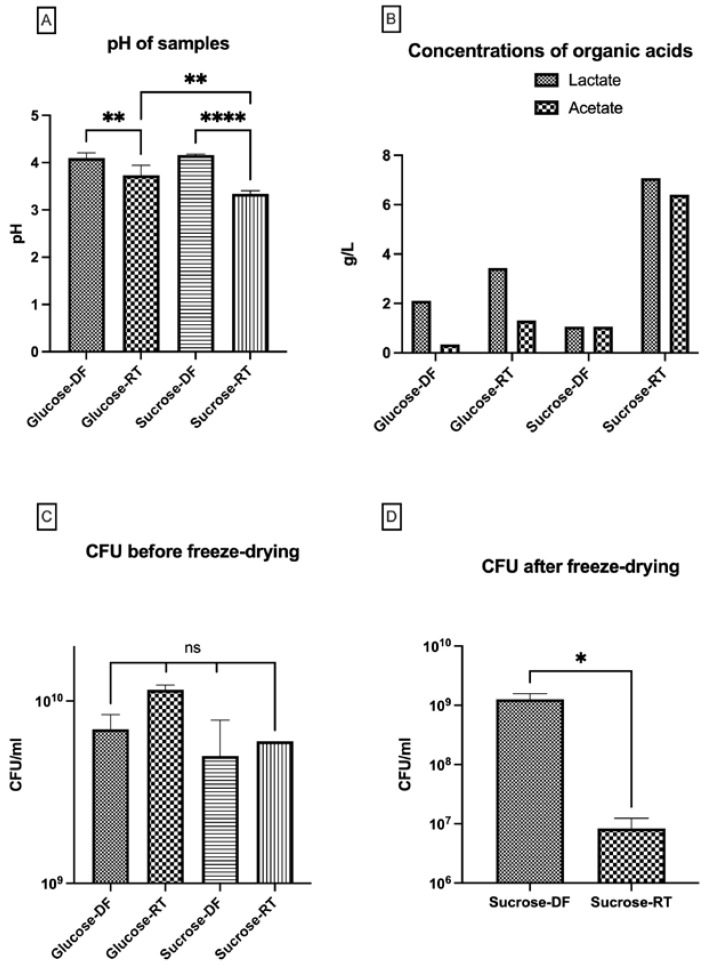
The effect of lyoconversion on pH (**A**), concentrations of organic acids (**B**), CFU after formulation and freezing (**C**), and CFU after freeze-drying (**D**). Samples incubated at room temperature overnight are denoted as RT, and direct-frozen samples are denoted as DF. Analysis of organic acids by HPLC indicated metabolic activity during the lyoconversion, while the concentrations of lactate and acetate increased substantially in the RT samples, especially for sucrose-grown bacteria. Significance levels used were * *p* < 0.05; ** *p* < 0.01; **** *p* < 0.0001, ns *p* > 0.05. Prism GraphPad version 9.0 was used for statistical analysis.

**Figure 7 microorganisms-12-02058-f007:**
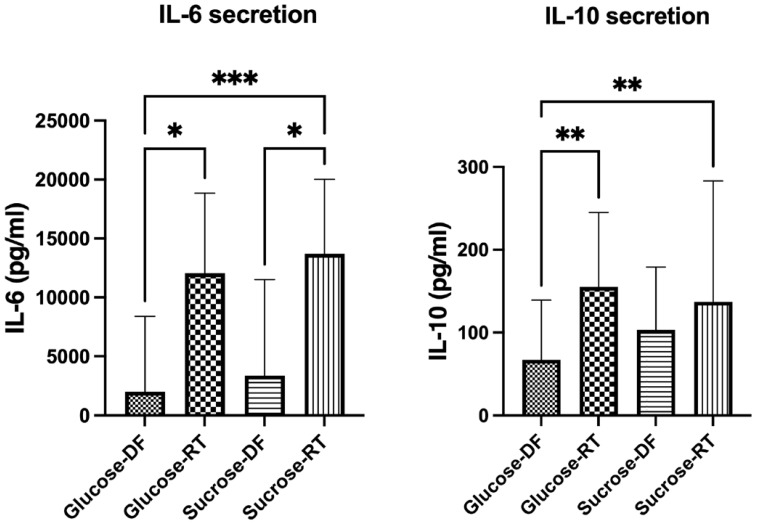
Induction of IL-6 and IL-10 by the different types of samples was measured in unstimulated PBMCs. Cytokine secretion was significantly increased in response to lyoconverted (RT) samples compared to direct-frozen (DF) samples. The multiplicity of bacteria equivalents was 10 for IL-6 secretion and 100 for IL-10 secretion. Friedman’s test was used for statistical analysis. Significance levels used were * *p* < 0.05; ** *p* < 0.01; *** *p* < 0.001. Prism GraphPad version 9.0 was used for statistical analysis.

**Figure 8 microorganisms-12-02058-f008:**
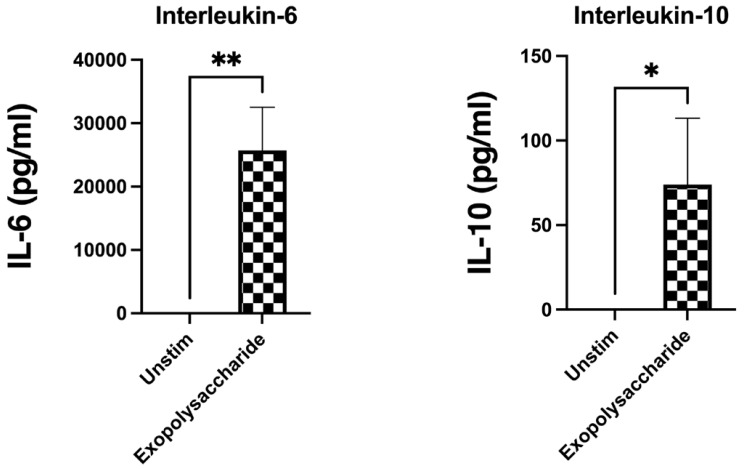
IL-6 and IL-10 induction in PBMCs after incubation with purified EPSs from sucrose-grown lyoconverted samples. Both IL-6 and IL-10 levels were significantly increased compared to unstimulated immune cells. The multiplicity of bacterial equivalent was 100. Wilcoxon matched-pair signed-rank test was used for statistical analysis. Significance levels used were * *p* < 0.05; ** *p* < 0.01. Prism GraphPad version 9.0 was used for statistical analysis.

## Data Availability

Data is available at the Open Science Framework database: https://osf.io/kfm5z/?view_only=010271aae5f14142b2ac9f57b7e2f8fd.

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
