# Peer review of "Limosilactobacillus reuteri DSM 17938 Produce Bioactive Components during Formulation in Sucrose"

_microorganisms, 2024, doi:10.3390/microorganisms12102058_

Round 1
Reviewer 1 Report
Comments and Suggestions for Authors
The manuscript investigated the effect of sucrose, used as a cytoprotectant, on the bioactive substances produced by Limosilactobacillus reuteri DSM 17938 during the freeze-drying process. The authors showed that sucrose was metabolized by the strain to produce a glucan-type exopolysaccharide (EPS), and other potential bioactive compounds, such as tryptamine and glycerol. The synthesized substances, especially EPS, induced the secretion of IL-6 and IL-10. The manuscript is well written and most of the results are supported by the data. The manuscript requires only minor revisions prior to publication. Other comments are shown below.
1. L221, where is the signal of fructose since fructose was also released during the lyoconversion.
2. L233, please provide data about the chemical shifts in a convenient manner for comparison.
3. The data are not visible in Figure 4, Figure 6, Figure 7 and Figure 8.
4. To fully evaluate the bioactivity of the lyoconverted substances, other cytokines should be examined.
Author Response
Comment 1
L221, where is the signal of fructose since fructose was also released during the lyoconversion.
Response 1
Signals from fructose could not be detected in the NMR spectra (including 1D 1H, 2D COSY, TOCSY, HSQC or HSQC-TOCSY).
Comment 2
L233, please provide data about the chemical shifts in a convenient manner for comparison.
Response 2
We thank the reviewer for this suggestion and the chemicals shifts are now listed as a table in the Supplementary material (Supplementary Table 3). We also pasted the table below.
Supplementary Table 3: 1H and 13C NMR chemical shifts (d, ppm) for the resonances from the a-glucan in the sucrose-RT and glucose-RT samples
|
H1/C1 |
H2/C2 |
H3/C3 |
H4/C4 |
H5/C5 |
H67C6 |
|
|
|
|
|
|
|
→ 4)-a-D-Glc |
5,34, 5.33, 5.30 102.5 |
3.59 74.2
|
4.05 75.9 |
3.65 80.3 |
3.98 72.5 |
3.72. 3.84* 63.6 |
→ 6)-a-D-Glc |
(a)4.95, 4.94 (b)4.94, 4.95, 4,96 100.9 |
3.56 74.4 |
3.73 76.2 |
3.45 c |
3.89 c |
c |
- 1H chemical shifts of anomeric protons in the sucrose-RT sample
- 1H chemical shifts of anomeric protons in the glucose-RT sample
- Not assigned unambiguously
Due to spectral overlap and line broadening, it was not possible to obtain specific assignments of the different structural fragments with the exception for the anomeric protons.
Comment 3
The data are not visible in Figure 4, Figure 6, Figure 7 and Figure 8.
Response 3
We thank the reviewer for pointing this out. It appears there had been some formatting error in the Word to PDF conversion. This has been adjusted.
Comment 4
To fully evaluate the bioactivity of the lyoconverted substances, other cytokines should be examined.
Response 4
We agree with the reviewer that a broader panel of cytokines and ratio of cytokines would be interesting to evaluate. This is something that we plan on performing for a continuation of the bioactivity-mapping of lyoconversion.
Reviewer 2 Report
Comments and Suggestions for Authors
This paper is exceptionally well-written, and its content is meaningful. It represents a noteworthy research achievement that deserves publication. However, a number of points need clarifying and certain statements require further justification. There are given below.
1. In the abstract section, the author explicitly outlines the significance of fermentation parameters in the practical production of probiotics, which leads to the exploration of freeze-drying protectants and the presentation of the core concept of this research. Through various experiments, the study examines different metabolites generated by the strain during its metabolic process when sucrose is utilized, approaching this from multiple angles. However, the logical connections between certain experiments and their operational objectives remain unclear. For instance, it is unclear what the logical relationship is between the immune regulation models throughout the experiment. In the formulation step, bacterial metabolic activity can serve as a means to modify bacterial viability, potentially enhancing the effectiveness of probiotics. This conclusion bears no progression or causal relationship with the preceding content and offers no practical guidance for the actual fermentation process.
2. In the introduction section, the author mentioned that Limosilactobacillus reuteri DSM 17938 can metabolize glucanase and other products from sucrose. This part of the background introduction was too simple and did not provide detailed background information.
3. The introduction of research methods was simple and clear.In the experiments with activated strains, medium sterilization conditions were not involved ;In 2.2, the freeze-drying conditions are not clearly stated.The overall structure of the article was messy, the arrangement of charts was not regular, and the format was not uniform.
4. The authors did not articulate the limitations of the study, although there was a complete analysis of the results, did not excite the reader about the research prospects.
5. The discussion part is too long, too much preparation for the conclusion is drawn,lines 348 to 371 can be simplified, and the research introduction of freeze-drying protectants can be placed in the introduction. The whole experimental design simply lists the different data measured by various experimental methods, and there is no logical relationship between experiments. The introduction of the experiment in the immunomodulatory model was also very abrupt, the experiment was not targeted, and the reason for the selected measure was not clearly introduced.
6. The title covers a wide and general range, does not reflect the central idea of the article, and looks like the title of a review.The question does not refer to freeze-drying or freeze-protectants in the fermentation process.The effect of its bioactive components on strain metabolism should be added at the end.
Author Response
Comment 1
In the abstract section, the author explicitly outlines the significance of fermentation parameters in the practical production of probiotics, which leads to the exploration of freeze-drying protectants and the presentation of the core concept of this research. Through various experiments, the study examines different metabolites generated by the strain during its metabolic process when sucrose is utilized, approaching this from multiple angles. However, the logical connections between certain experiments and their operational objectives remain unclear. For instance, it is unclear what the logical relationship is between the immune regulation models throughout the experiment. In the formulation step, bacterial metabolic activity can serve as a means to modify bacterial viability, potentially enhancing the effectiveness of probiotics. This conclusion bears no progression or causal relationship with the preceding content and offers no practical guidance for the actual fermentation process.
Response 1
We have changed the abstract section slightly according to this suggestion to better display the reasoning behind the immunomodulatory experiments and thank the reviewer for this helpful comment (line 26-27).
Regarding “In the formulation step, bacterial metabolic activity can serve as a means to modify bacterial viability, potentially enhancing the effectiveness of probiotics. This conclusion bears no progression or causal relationship with the preceding content and offers no practical guidance for the actual fermentation process.”: In the abstract we are not saying that bacterial metabolic activity can serve as a means to modify bacterial viability, but rather that “we have shown that metabolic activities of the bacteria during the formulation step can be used as a tool to alter the activity of the bacteria, and thereby potentially improve probiotic efficacy.”. The purpose of this statement is to showcase the potential intervention window that lyoconversion constitutes when attempting to increase the bioactivity of probiotics.
Comment 2
In the introduction section, the author mentioned that Limosilactobacillus reuteri DSM 17938 can metabolize glucanase and other products from sucrose. This part of the background introduction was too simple and did not provide detailed background information. Response 2
We thank the reviewer for this comment. The glucan that are produced by L. reuteri DSM 17938 has been studied extensively by others, and we decided to add an additional reference (line 73). The cited paper deals with the mother strain of DSM 17938 and its exopolysaccharide production during growth on different sugars. However, the production of EPS and other compounds in simple sugar solutions (such as the freeze-drying formulation in this paper) has not been described elsewhere, and hence we are not including any further information on this topic in the introduction but would rather emphasize this as a finding of this paper.
Comment 3
The introduction of research methods was simple and clear. In the experiments with activated strains, medium sterilization conditions were not involved ;In 2.2, the freeze-drying conditions are not clearly stated. The overall structure of the article was messy, the arrangement of charts was not regular, and the format was not uniform.
Response 3
Medium sterilization has been added to the methods section (line 81).
We have added the freeze-drying equipment used in section 2.2, which is the same as in the EPS extraction method, (Coolsafe freeze dryer) which doesn’t have any program but rather dries samples using a condensation-trap and a vacuum pump (line 103).
We thank the reviewer for this comment, and we have changed most figures and charts, and unified the text formatting.
Comment 4
The authors did not articulate the limitations of the study, although there was a complete analysis of the results, did not excite the reader about the research prospects.
Response 4
We have added information about the use within postbiotics (line 382-385) and thank the reviewer for this comment.
Comment 5
The discussion part is too long, too much preparation for the conclusion is drawn, lines 348 to 371 can be simplified, and the research introduction of freeze-drying protectants can be placed in the introduction. The whole experimental design simply lists the different data measured by various experimental methods, and there is no logical relationship between experiments. The introduction of the experiment in the immunomodulatory model was also very abrupt, the experiment was not targeted, and the reason for the selected measure was not clearly introduced.
Response 5
We thank the reviewer for this comment and have reduced and simplified the mentioned section, now line 353 – 370.
We have further clarified why we evaluated the converted samples in a model for immune modulation in the discussion and reasoning behind the selection of IL-6 and IL-10 was added in the results section (line 322-324).
Comment 6
The title covers a wide and general range, does not reflect the central idea of the article, and looks like the title of a review. The question does not refer to freeze-drying or freeze-protectants in the fermentation process. The effect of its bioactive components on strain metabolism should be added at the end.
Response 6
We believe that the title is descriptive of the content of the manuscript and think it is appropriate.
Reviewer 3 Report
Comments and Suggestions for Authors
While the topic of this paper is interesting and relevant, the article requires significant improvements. The clarity, structure, and connection between the results and conclusions need to be strengthened to better convey the value of the research. My observations are listed below:
The references are not cited correctly in the text and do not comply with the guidelines of the journal Microorganisms, the authors should correct this aspect – Moreover, there are multiple fonts in the text – please check and correct them."
The introduction provides no comprehensive background or justification for the study, Please expand the introduction by providing a more thorough review of recent literature on probiotics, lyoconversion, and bioactive components.
The purpose of the study should be explicitly stated, so the reader understands what the authors are trying to achieve. The purpose of the article is missing from the introduction – The authors must introduce the aim of the article
The results section presents a wealth of data, but it is poorly organized…
For Figure 1 if possible, the authors should use a higher resolution
The paper presents a large amount of data (e.g., NMR, HPLC, and GC-MS results), but there is insufficient explanation for how these data contribute to the overall conclusions. For example, the NMR data in Figures 2-4 are detailed, but their biological relevance is not adequately discussed.
Please add the meaning of A,B,C,D on Figure 6
The results section fails to critically assess some findings. For example, while it is mentioned that lyoconversion reduces freeze-drying survival, no in-depth analysis is offered on how this could impact the application of probiotics in real-world conditions.
The production of tryptamine and glycerol is mentioned, but there is no detailed discussion of their roles in the context of probiotic efficacy or health outcomes. Have they any roles?
Please ensure that all claims are supported by citations that follow the journal’s guidelines.
Although the study claims to present novel findings (such as the concept of "lyoconversion"), it does not clearly emphasize what makes these findings unique or impactful. Please clearly highlight the novel aspects of the findings.
For example, emphasize how the discovery of lyoconversion as a process that produces bioactive metabolites is a significant contribution to the field of probiotic research.
The conclusion is brief and does not effectively summarize the key findings or propose meaningful next steps for research.
The article seems to have been written in a hurry, carelessly
Author Response
Comment 1
The references are not cited correctly in the text and do not comply with the guidelines of the journal Microorganisms, the authors should correct this aspect – Moreover, there are multiple fonts in the text – please check and correct them."
Response 1
There was a formatting mistake when transferring the content of the paper to the MDPI microorganisms template, and this has been corrected. We thank the reviewer for noticing.
We have also adjusted the references to fit the journal guidelines. We did however not do that when we submitted the paper as the journal allows free-format submission.
Comment 2
The introduction provides no comprehensive background or justification for the study, Please expand the introduction by providing a more thorough review of recent literature on probiotics, lyoconversion, and bioactive components.
Response 2
We have scrutinized the introduction and believe that the section beginning with “Industrial production of probiotics is accompanied by several challenges and differs substantially from lab-scale production”, line 65 and forward, provides justification for the study.
- We have scrutinized the introduction and believe that the section beginning with “Industrial production of probiotics is accompanied by several challenges and differs substantially from lab-scale production”, line 65 and forward, provides justification for the study. However, we have clarified the purpose of the study in the end of the introduction (line 73-77).
Since this paper doesn’t deal with probiotics in general, but only the subpopulation of probiotic species that have an active glucansucrase enzyme, we would not like to expand the introduction with general probiotics information. To our knowledge, lyoconversion has not been described before and hence, there are no references known to us. Bioactive components are mentioned in the form of EPS, and while there many other compounds that were produced during lyoconversion, none of them were tested specifically and therefore we would not like to include broad information on bioactives.
Comment 3
The purpose of the study should be explicitly stated, so the reader understands what the authors are trying to achieve. The purpose of the article is missing from the introduction – The authors must introduce the aim of the article.
Response 3
We thank the reviewer for this comment and have added a clear purpose of the study (line 73-77).
Comment 4
The results section presents a wealth of data, but it is poorly organized…
For Figure 1 if possible, the authors should use a higher resolution
Response 4
We have improved the quality of the image and thank the reviewer for this comment.
Comment 5
The paper presents a large amount of data (e.g., NMR, HPLC, and GC-MS results), but there is insufficient explanation for how these data contribute to the overall conclusions. For example, the NMR data in Figures 2-4 are detailed, but their biological relevance is not adequately discussed.
Response 5
We agree with the reveiwer that we have been stringent on the discussion of the biological relevance of the compounds detected. We decided to avoid talking about the meaning as we have no absolute, but only relative, quantification of many of the compounds (the ones detected by GCMS) and hence, it cannot be said for sure that the concentration are high enough to elicit a bioactive effect. However, we did select some of the detected compounds which we discussed as we belive these may have biological importace (e.g. glycerol, tryptamine), especially if the lyoconversion can be increased. Regarding NMR-data in Figure 2-3 demonstrates that sucrose is converted into EPS, and the biological relevance of EPS is shown in the immunomodulatory experiments. It is also discussed at lines 386-396.
Comment 6
Please add the meaning of A,B,C,D on Figure 6
Response 6
We thank the reviewer for noticing the lack of explanatory letters in the figure legend, which has now been added.
Comment 7
The results section fails to critically assess some findings. For example, while it is mentioned that lyoconversion reduces freeze-drying survival, no in-depth analysis is offered on how this could impact the application of probiotics in real-world conditions.
Response 7
We believe that we have discussed the topic of freeze-drying survival in the second section of the discussion (line 371-385). There, we discuss the consumption of the non-reducing disaccharide sucrose and increasing Maillard reactions. We also provide a suggestion on how the stability-associated issues with lyoconversion may be relieved by an additional addition of sucrose. Regarding real-world conditions, a producer normally aims at reaching high levels of freeze drying survival (>50%), and we show that lyoconversion freeze-drying survival is very low (0.14%). We have added the freeze-drying survival calculations in the results section (line 311-312).
Comment 8
The production of tryptamine and glycerol is mentioned, but there is no detailed discussion of their roles in the context of probiotic efficacy or health outcomes. Have they any roles?
Response 8
Both tryptamine and glycerol and their bioactivities are discussed at line 409 – 429.
Comment 9
Please ensure that all claims are supported by citations that follow the journal’s guidelines.
Response 9
The reference-section has been adjusted according to the journal guidelines and thank the reviewer for noticing.
Comment 10
Although the study claims to present novel findings (such as the concept of "lyoconversion"), it does not clearly emphasize what makes these findings unique or impactful. Please clearly highlight the novel aspects of the findings. For example, emphasize how the discovery of lyoconversion as a process that produces bioactive metabolites is a significant contribution to the field of probiotic research.
Response 10
We thank the reviewer for this question. While the concept of lyoconversion is being described in this paper, we recognize that the phenomenon has not been studied enough to draw any firm conclusions as to what this may mean for the probiotic industry, and in order to avoid excess speculation, we did not discuss how the concept may impact the field. Nevertheless, the problem with survival needs to be solved in order for lyoconversion to relevant for the industry. However, we have added another aspect, namely how it may influence and improve the efficacy of postbiotics (Line 378-385).
Comment 11
The conclusion is brief and does not effectively summarize the key findings or propose meaningful next steps for research.
Response 11
We thank the reviewer for this suggestion and have added suggestions on future research directions to the conclusion (Line 435-437).
Round 2
Reviewer 2 Report
Comments and Suggestions for Authors
The author already reviewed the comments accordingly
Author Response
Comment
The author already reviewed the comments accordingly
Response
We thank the reviewer for their valuable comments and suggestions.
Reviewer 3 Report
Comments and Suggestions for Authors
The article still not meet the guidelines of the journal – especially in terms of writing the bibliography
please see https://www.mdpi.com/journal/microorganisms/instructions
It is true that the journal accepts allows free-format submission, but when your manuscript reaches the revision stage, you will be requested to format the manuscript according to the journal guidelines.
there are different fonts in the text .. see the table from Figure 1
I understood the opinion of the authors regarding the introduction, however I believe that it could be better written, bringing more arguments for glucansucrase enzyme, lyoconversion novelty..
Regarding "lyoconversion," even though this concept hasn’t been described before, it would be useful to mention the context of similar processes or explain what makes "lyoconversion" different and innovative.
The discussion about the biological relevance of compounds detected through NMR and GC-MS is limited due to the lack of absolute quantification. This leads to an avoidance of an in-depth discussion of the implications of these compounds for human health.
Even if they do not have absolute quantification, the authors could suggest future experiments that would more precisely quantify these compounds to bring more insight into their biological relevance.
The discussion about the impact of "lyoconversion" on freeze-drying survival is general, and the proposed solutions (such as sucrose supplementation) are handled superficially.
It would be helpful to mention similar examples from the literature where other methods were used to enhance bacterial survival during freeze-drying and discuss if those methods could be applicable for "lyoconversion" as well.
Author Response
Comment 1
The article still not meet the guidelines of the journal – especially in terms of writing the bibliography. Please see https://www.mdpi.com/journal/microorganisms/instructions. It is true that the journal accepts allows free-format submission, but when your manuscript reaches the revision stage, you will be requested to format the manuscript according to the journal guidelines.
Response 1
We have adjusted the Bibliography and thank the reviewer for this comment.
Comment 2
There are different fonts in the text .. see the table from Figure 1.
Response 2
We thank the reviewer for noticing and have adjusted the font in Figure 1.
Comment 3
I understood the opinion of the authors regarding the introduction, however I believe that it could be better written, bringing more arguments for glucansucrase enzyme, lyoconversion novelty.
Response 3
We have revised the introduction at line 71-80 where we have added more information regarding the novelty of lyoconversion.
Comment 4
Regarding "lyoconversion," even though this concept hasn’t been described before, it would be useful to mention the context of similar processes or explain what makes "lyoconversion" different and innovative.
Response 4
We have added additional reasoning regarding the lyconversion and its novelty and innovative strength to the discussion and thank the reviewer for the suggestion (line 373-375).
Regarding simliar processes, we have investigated the literature regarding use of holding times during production of probiotics and are unaware of any such studies.
Comment 5
The discussion about the biological relevance of compounds detected through NMR and GC-MS is limited due to the lack of absolute quantification. This leads to an avoidance of an in-depth discussion of the implications of these compounds for human health. Even if they do not have absolute quantification, the authors could suggest future experiments that would more precisely quantify these compounds to bring more insight into their biological relevance.
Response 5
We agree that it would be good to emphasize that absolute quantification is warranted and have added this into the discussion at line 441-445.
Comment 6
The discussion about the impact of "lyoconversion" on freeze-drying survival is general, and the proposed solutions (such as sucrose supplementation) are handled superficially. It would be helpful to mention similar examples from the literature where other methods were used to enhance bacterial survival during freeze-drying and discuss if those methods could be applicable for "lyoconversion" as well.
Response 6
We have added some more suggestions and alterations that may partly rescue or solve the issue with freeze-drying survival (line 387-396). We also added two references for the suggested methodologies.